# Question Decomposition with Dependency Graphs

**Matan Hasson**                                             MATANHASSON@MAIL.TAU.AC.IL
*Tel Aviv University*
**Jonathan Berant**                                             JOBERANT@CS.TAU.AC.IL
*Tel Aviv University, The Allen Institute for AI*

## Abstract

QDMR is a meaning representation for complex questions, which decomposes questions into a sequence of atomic steps, and has been recently shown to be useful for question answering. While state-of-the-art QDMR parsers use the common sequence-to-sequence (seq2seq) approach, a QDMR structure fundamentally describes labeled relations between spans in the input question, and thus dependency-based approaches seem appropriate for this task. In this work, we present a QDMR parser that is based on *dependency graphs (DGs)*, where nodes in the graph are words and edges describe logical relations that correspond to the different computation steps. We propose (a) a non-autoregressive graph parser, where all graph edges are computed simultaneously, and (b) a seq2seq parser that uses the gold graph as auxiliary supervision. We find that a graph parser leads to a moderate reduction in performance (0.47→0.44), but to a 16x speed-up in inference time due to its non-autoregressive nature, and to improved sample complexity compared to a seq2seq model. Second, training a seq2seq model with auxiliary DG supervision leads to better generalization on out-of-domain data and on QDMR structures with long sequences of computation steps.

## 1. Introduction

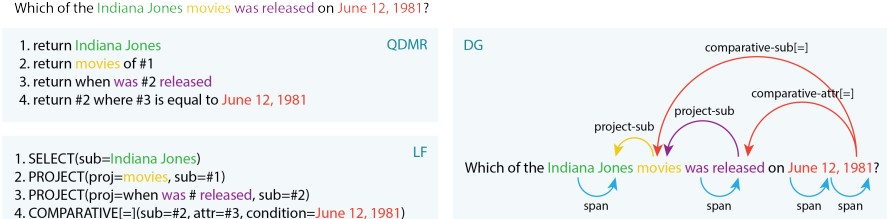

Figure 1: An example question with its corresponding QDMR structure (top-left), dependency graph over the question tokens (right), and an intermediate logical form (LF) used for the QDMR→DG conversion and for evaluation.

Training neural networks to reason over multiple parts of their inputs across modalities such as text, tables, and images, has been a focal point of interest in recent years [Antol et al., 2015, Pasupat and Liang, 2015, Johnson et al., 2017, Suhr et al., 2019, Welbl et al., 2018, Talmor and Berant, 2018, Yang et al., 2018, Hudson and Manning, 2019, Dua et al., 2019, Chen et al., 2020, Hannan et al., 2020, Talmor et al., 2021]. The most common way to

probe whether a model is capable of complex reasoning, is to pose in natural language a complex question, which requires performing multiple steps of computation over the input.

One natural way of answering such complex questions is to break them down into a sequence of simpler sub-steps [Christiano et al., 2018, Min et al., 2019, Perez et al., 2020]. Wolfson et al. [2020] recently proposed QDMR, a meaning representation where complex questions are represented through a sequence of simpler atomic executable steps (see Fig. 1), and the final answer is the answer to the final step. QDMR has been shown to be useful for multi-hop question answering (QA) [Wolfson et al., 2020] and also for improving interpretability in visual QA [Subramanian et al., 2020].

State-of-the-art QDMR parsers use the typical seq2seq approach. However, it is natural to think of QDMR as a dependency graph over the input question tokens. Consider the example in Fig. 1. The first QDMR step selects the span *"Indiana Jones"*. Then, the next step uses a PROJECT operation to find the *"movies"* of Indiana Jones, and the next step uses another PROJECT operation to find the date when the movies were *"released"*. Such relations can be represented as labeled edges over the relevant question tokens.

In this work, we propose to use the dependency graph view of QDMR to improve QDMR parsing. We describe a conversion procedure that automatically maps QDMR structures into dependency graphs, using a structured intermediate logical form representation (Fig 1, bottom-left). Once we have graph supervision for every example, we train a dependency graph parser, in the spirit of Dozat and Manning [2018], where we predict a labeled relation for every pair of question tokens, representing the logical relation between the tokens. Unlike seq2seq models, this is a non-autoregressive parser, which decodes the entire output structure in a single step.

In addition, we study the effect of using dependency graphs as auxiliary supervision for a seq2seq QDMR parser, where the graph is decoded from the encoder representations. Towards that end, we propose a Latent-RAT encoder, which uses relation-aware transformer [Shaw et al., 2018] to explicitly represent the relation between every pair of input tokens. Relation-aware transformer has been shown to be useful for encoding graph structures in the context of semantic parsing [Wang et al., 2020].

Last, to fairly compare QDMR parsers that use different representations, we propose an evaluation metric, LF-EM, based on the aforementioned intermediate logical form. We show that LF-EM correlates better with human judgements compared to existing metrics.

We find that our graph parser leads to a small reduction in LF-EM compared to seq2seq models ($0.47{\to}0.44$), but is 16x faster due to its non-autoregressive nature, and is by design more interpretable. Moreover, it has better sample complexity and outperforms the seq2seq model when trained on 10% of the data or less. When training a seq2seq model with the auxiliary graph supervision, the parser obtains similar performance as when trained on the entire dataset (0.471 LF-EM), but substantially improves performance when generalizing to new domains. Moreover, it performs better on examples with a large number of computation steps. Our code is available at https://github.com/matanhasson/qdecomp_with_dependency_graphs.

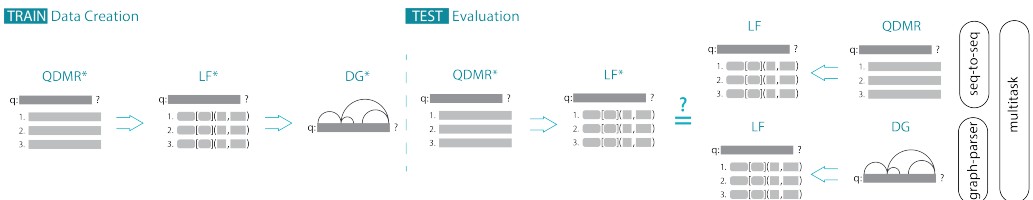

Figure 2: Overview. For training (left), we create gold DGs from gold QDMRs (§4) through a conversion into LFs (§3). At test time (right), we convert model predictions, either QDMRs or DGs, into LFs (§3, §4), and evaluate by comparing them to the gold LFs. Asterisk (*) denotes gold representations.

## 2. Overview

The core of this work is to examine the utility of a dependency graph (DG) representation for QDMR. We propose conversion procedures that enable training and evaluating with DGs (see Fig. 2). First, we convert gold QDMR structures into logical forms (LF), where each computation step in QDMR is represented with a formal operator, properties and arguments (§3). Then, we obtain gold DGs by projecting the logical forms onto the question tokens (§4). Once we have question- DG pairs, we can train a graph parser. At test time, QDMRs and DGs are converted into LFs for evaluation. We propose a new evaluation metric over LFs (§3), and show it is more robust to semantic-preserving changes compared to prior metrics.

Our proposed parsers are in §5. On top of standard seq2seq models, we describe (a) a graph parser, and (b) a multi-task model, where the encoder of the seq2seq model is also trained to predict the DG.

## 3. QDMR Logical Forms

QDMR [Wolfson et al., 2020] is a text-based meaning representation focused on representing the meaning of complex questions. It is based on a decomposition of questions into a sequence of simpler executable *steps* (Fig. 1), where each step corresponds to a SQL-inspired operator (Table 5, §A.1). We briefly review QDMR and then define a logical form (LF) representation based on these operations. We use the LFs both for mapping QDMRs to DGs, and also to fairly evaluate the output of parsers that output either QDMRs directly or DGs.

**QDMR Definition**   Given a question with $n$ tokens, $q = q_1 \ldots q_n$, its QDMR is a sequence of $m$ steps $s^1, \ldots, s^m$, where step $s^i$ conceptually maps to a single logical operator $o^i \in \mathcal{O}$. A step, $s^i$, is a sequence of $n_i$ tokens $s^i = s^i_1 \ldots s^i_{n_i}$, where token $s^i_j$ is either a question token $\in \mathcal{V}_q$ (or some inflection of it), a word from a constant predefined lexicon $\in \mathcal{V}_{\text{const}}$, or a reference token $\in \mathcal{V}^i_{\text{ref}} = \{\#1, \ldots, \#(i-1)\}$, referring to a previous step. Fig. 3 shows an example for a question and its QDMR structure.

**QDMR Logical Form (LF)**   Given a QDMR $S = \langle q; s^1, \ldots, s^m \rangle$, its *logical form* is a sequence of logical form steps $Z = \langle q; z^1, \ldots, z^m \rangle$. The LF step $z^i$, corresponding to $s^i$, is a triplet $z^i = \langle o^i, \rho^i, A^i \rangle$ where $o^i \in \mathcal{O}$ is the logical operator; $\rho^i \in PROP_{o^i}$ are operator-specific properties; and $A^i$ is a dictionary of arguments, mapping an operator-specific argument

*"Which group from the census is smaller: Pacific islander or African American?"*

| | |
|---|---|
| 1. return census groups | $\mathcal{V}_q = \{\dots group, \textbf{groups}, \dots small, smaller, smallest, \dots\}$ |
| 2. return #1 that is Pacific islander | $\mathcal{V}_{\text{ref}}^5 = \{\#1, \#2, \textbf{\#3}, \#4\}$ |
| 3. return #1 that is African American | |
| 4. return size of #2 | $\mathcal{V}_{\text{const}} = \mathcal{V}_{op} \cup \mathcal{V}_{store} \cup \mathcal{V}_{aux}$ |
| 5. return size of #3 | $\mathcal{V}_{op} = \{difference, sum, \textbf{lowest}, highest, for each, \dots\}$ |
| 6. return which is lowest of #4 , #5 | $\mathcal{V}_{store} = \{population, \textbf{size}, elevation, flights, price, date \dots\}$ |
| | $\mathcal{V}_{aux} = \{a, is, are, \textbf{of}, \textbf{that}, the, with, was, did, to \dots\}$ |

Figure 3: QDMR annotation vocabularies. Each example is annotated with a lexicon that consists of: $\mathcal{V}_q$, the question tokens and their inflections; $\mathcal{V}_{\text{ref}}^i$, references to previous steps; $\mathcal{V}_{\text{const}}$, constant terms including operational terms ($\mathcal{V}_{op}$), domain-specific words that are not in the question, such as *size* ($\mathcal{V}_{store}$); and auxiliary words like prepositions ($\mathcal{V}_{aux}$). Boldface indicates words used in the QDMR structure.

| Operator | PROP | ARG | Example |
|---|---|---|---|
| SELECT | $\emptyset$ | sub | return cubes |
| | | | SELECT[](sub=cubes) |
| FILTER | $\emptyset$ | sub, cond | return #1 from Toronto |
| | | | FILTER[](sub=#1, cond=from Toronto) |
| AGGREGATE | *max, min, count,* | arg | return maximal number of #1 |
| | *sum, avg* | | AGGREGATE[*max*](arg=#1) |
| ARITHMETIC | *sum, diff, mult, div* | arg, left, right | return the difference of #3 and #4 |
| | | | ARITHMETIC[*diff*]( left=#3, right=#4) |

Table 1: LF operators, properties and arguments (partial list, see Table 5 for full list).

$\eta \in ARG_{o^i, \rho^i}$ to a span $\tau$ from the QDMR step $s^i$. For convenience, we denote $z^i$ with the string $o^i[\rho^i](\eta_1^i = \tau_1^i, \dots)$. Table 1 provides a few examples for the mapping from QDMR to LF steps, and Table 5 (§A.1) provides the full list.

**QDMR→LF** We convert QDMRs to LFs with a rule-based method, extending the procedure for detecting operators from Wolfson et al. [2020] to also find properties and arguments. To detect properties we use a lexicon (see Table 6 in §A.1).

**LF-based Evaluation (LF-EM)** The official evaluation metric for QDMR[1] is normalized EM (NormEM), where the predicted and gold QDMR structures are normalized using a rule-based procedure, and then exact string match is computed between the two normalized QDMRs. Since in this work we convert both QDMRs (§3) and DGs (§4) to LFs, we propose a LF-based evaluation metric.

LF-EM essentially involves computing exact match between the predicted and gold LFs. To further capture semantic equivalences, we perform more normalization steps, which for brevity are described in §A.2. We manually evaluate the metrics NormEM and LF-EM on 50 random development set examples using predictions from the COPYNET+BERT model (see §6). We find that both metrics have perfect precision (no false-positives); but the LF-EM covers more examples (52.0% vs 40.0%). Thus, it provides a tighter lower bound on the performance of a QDMR parser and correlates better with notions of semantic equivalence.

---

1. https://leaderboard.allenai.org/break/submissions/public

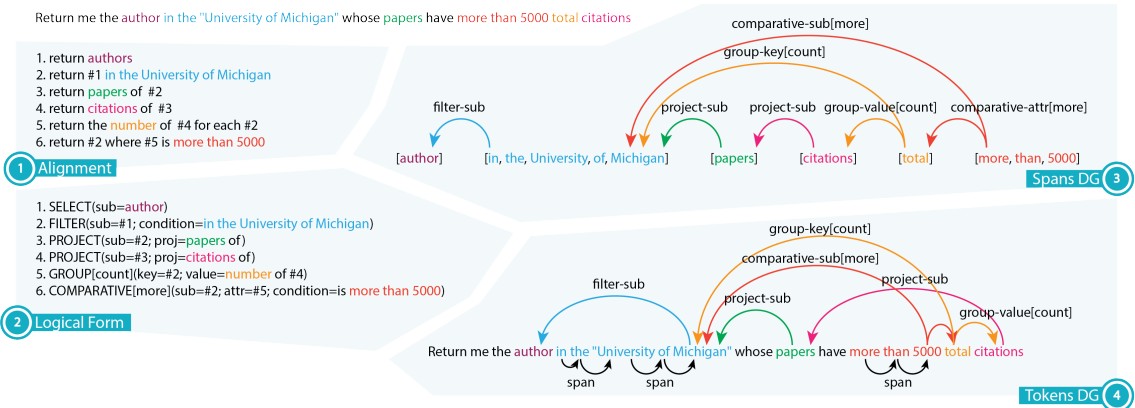

Figure 4: Dependency graph creation: (1) *Token alignment*: align question tokens and QDMR step tokens. (2) *Logical Form*: extract the LF of the QDMR. (3) *SDG extraction*: induced from the LF and the alignment. (4) *DG Creation*: convert the SDG to a DG.

## 4. From LFs to Dependency Graphs

Given a QDMR decomposition $S = \langle q; s^1, \ldots, s^m \rangle$, we construct a dependency graph $G = \langle \mathcal{N}, \mathcal{E} \rangle$, where the nodes $\mathcal{N}$ correspond to question tokens, and the edges $\mathcal{E}$ describe the logical operations, resulting in a graph with the same meaning as $S$.

The LF→DG procedure is shown in Fig. 4 and consists of the following steps:

- Token alignment: align each token in the question to a token in a QDMR step (§A.3).
- Spans Dependency Graph (SDG) extraction: construct a graph where each node corresponds to a list of tokens in a QDMR step, and edges describe the dependencies between the steps (§A.4).
- Dependency Graph (DG) extraction: convert the SDG to a DG over the question tokens. Here, we add `span` edges for tokens that are in the same step, and deal with some representation issues (§A.5).

Because we convert predicted DGs to LFs for evaluation, the LF→DG conversion must be invertible. Our conversion succeeds in 97.12% of the BREAK dataset [Wolfson et al., 2020].

## 5. Models

Once we have methods to convert QDMRs to DGs and LFs, and DGs to LFs, we can evaluate the advantages and disadvantages of standard autoregressive decoders compared to graph-based parsers. We describe three models: (a) An autoregressive parser, (b) a graph parser, (c) an autoregressive parser that is trained jointly with a graph parser in a multi-task setup. For a fair comparison, all models have the same BERT-based encoder [Devlin et al., 2019].

**CopyNet+BERT (baseline)**    This autoregressive QDMR parser is based on the COPY-NET baseline from Wolfson et al. [2020], except we replace the BiLSTM encoder with a

transformer initialized with BERT. The model encodes the question $q$ and then decodes the QDMR $S$ step-by-step and token-by-token. The decoder is an LSTM [Hochreiter and Schmidhuber, 1997] augmented with a copy mechanism [Gu et al., 2016], where at each time step the model either decodes a token from the vocabulary or a token from the input. Since the input is tokenized with word pieces, we average word pieces that belong to a single word to get word representations, which enables word copying. Training is done with standard maximum likelihood.

**Biaffine Graph Parser (BiaffineGP)** The biaffine graph parser takes as input the question $q$ augmented with the special tokens described in §A.5 and predicts the DG by classifying for every pair of tokens whether there is an edge between them and the label of the edge. The model is based on the biaffine dependency parser of Dozat and Manning [2018], except here we predict a $DAG$ and not a tree, so each node can have more than one outgoing edge.

Let $\mathbf{H} = \langle \mathbf{h}_1, \ldots, \mathbf{h}_{|\mathbf{H}|} \rangle$ be the sequence of representations output by the BERT encoder concatenated with the POS embeddings. The biaffine parser uses four 1-hidden layer feed-forward networks over each contextualized token representation $\mathbf{h}_t$:

$$\mathbf{h}_t^{\text{edge-head}} = FF^{\text{edge-head}}(\mathbf{h}_t), \mathbf{h}_t^{\text{edge-dep}} = FF^{\text{edge-dep}}(\mathbf{h}_t),$$
$$\mathbf{h}_t^{\text{label-head}} = FF^{\text{label-head}}(\mathbf{h}_t), \mathbf{h}_t^{\text{label-dep}} = FF^{\text{label-dep}}(\mathbf{h}_t).$$

The probability of an edge from token $i$ to token $j$ is given by $\sigma(\mathbf{h}_i^{\text{edge-dep}^\top} W_{\text{edge}} \mathbf{h}_j^{\text{edge-head}})$, where $W_{\text{edge}}$ is a parameter matrix. Similarly, the score of an edge labeled by the label $l$ from token $i$ to token $j$ is given by $s_{ij}^l = \mathbf{h}_i^{\text{label-dep}^\top} W_l \mathbf{h}_j^{\text{label-head}}$, where $W_l$ is the parameter matrix for this label. We then compute a distribution over the set of labels $\mathcal{L}$ with $\text{softmax}(s_{ij}^1, \ldots, s_{ij}^{|\mathcal{L}|})$.

Training is done with maximum likelihood both on the edge probabilities and label probabilities. Inference is done by taking all edges with edge probability $> 0.5$ and then labeling those edges according to the most probable label.

There is no guarantee that the biaffine parser will output a valid DG. For example, if an SDG node has an outgoing edge labeled with `filter-sub` and another labeled with `project-sub`, we cannot tell if the operator is `FILTER` or `PROJECT`. This makes parsing fail, which occurs in 1.83% of the cases. To create a SDG, we first use the `span` edges to construct SDG nodes with lists of tokens, and then add edges between SDG nodes by projecting the edges between tokens to edges between the SDG nodes. To prevent cases where parsing fails, we can optionally apply an ILP that takes the predicted probabilities as input, and outputs a valid DG. The exact details are given in our open source implementation.

**Multi-task Latent-RAT Encoder (Latent-RAT)** In this model, our goal is to improve the seq2seq parser by providing more information to the encoder using the DG supervision. Our model will take the question $q$ (with special tokens as before) as input, and predict both the graph $G$ directly and the QDMR structure $S$ with a decoder.

We would like the information on relations between tokens to be part of the transformer encoder, so that the decoder can take advantage of it. To accomplish that, we use RAT transformer layers [Shaw et al., 2018, Wang et al., 2020], which explicitly represent relations

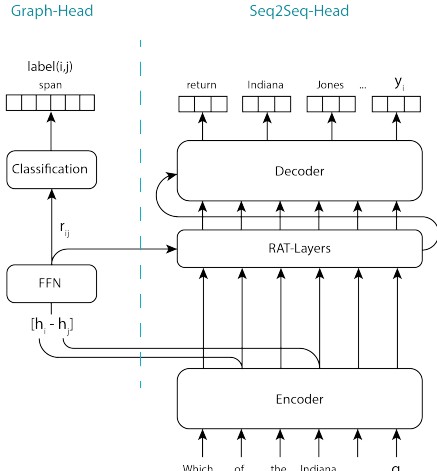

Figure 5: Latent-RAT architecture. The encoder hidden states represent the relations between the question tokens, $r_{ij}$. Then, these representations are used for (1) direct prediction of the dependency between the tokens (graph-head); (2) augment the encodings via RAT layers (seq2seq-head). The sub-networks for $r_{ij}^K$, $r_{i,j}^V$ are symmetric, and represented by the $r_{ij}$.

between tokens, and have been shown to be useful for encoding graphs over input tokens in the context of semantic parsing.

RAT layers inject information on the relation between tokens inside the transformer self-attention mechanism [Vaswani et al., 2017]. Specifically, the similarity score $e_{ij}$ computed using queries and keys is given by $e_{ij} \propto \mathbf{h}_i W_Q (\mathbf{h}_j W_K + r_{ij}^K)^T$, where $W_Q, W_K$ are the query and key parameter matrices and the only change is the term $r_{ij}^K$, which represents the relation between the tokens $i$ and $j$. Similarly, the relation between tokens is also considered when summing over self-attention values $\sum_{j=1}^{\mathbf{H}} \alpha_{ij}(x_j W_V + r_{ij}^V)$, where $W_V$ is the value parameter matrix, $\alpha_{ij}$ is the attention distribution and the only change is the term $r_{ij}^V$.

Unlike prior work where the terms $r_{ij}^K, r_{ij}^V$ were learned parameters, here we want these vectors to (a) be a function of the contextualized representation and (b) be informative for classifying the dependency label in the gold graph. By learning latent representations from which the gold graph can be decoded, we will provide useful information for the seq2seq decoder. Specifically, given the encoder output representations $\mathbf{h}_i, \mathbf{h}_j$ for tokens $i$ and $j$, we represent relations and compute a loss in each RAT layer as follows (Fig. 5):

$$
\begin{aligned}
r_{ij}^K &= FF^K(\mathbf{h}_i - \mathbf{h}_j), \\
S^K &= R^K W^{\text{out}} + b^K \in \mathbb{R}^{n \times n \times |\mathcal{L}|}, \\
Loss^K &= CE(S^K).
\end{aligned}
$$

$FF^K$ is a 1-hidden layer feed-forward network, $R^K \in \mathbb{R}^{(n \times n) \times d_{\text{transformer}}}$ is a concatenation of all $r_{ij}^K$ for all pairs of tokens, $W^{\text{out}} \in \mathbb{R}^{d_{\text{transformer}} \times |\mathcal{L}|}$ is a projection matrix that provides a score for all possible labels (including the NONE label).

We compute an analogous loss $Loss^V$ for $r_{ij}^V$ and the final graph loss is $Loss^K + Loss^V$ over all RAT layers. To summarize, by performing multi-task training with this graph loss we push the transformer to learn representations $r_{ij}$ that are informative of the gold graph, and can then be used by the decoder to output better QDMR structures.

## 6. Experiments

### 6.1 Experimental Setup

We build our models in AllenNLP [Gardner et al., 2018], and use BERT-base [Devlin et al., 2019] to produce contextualized token representations in all models. We train with the Adam optimizer [Kingma and Ba, 2015]. Our LATENT-RAT model includes 4 RAT layers, each with 8 heads. Full details on hyperparameters and training procedure in Appendix §A.7.

We examine the performance of our models in three setups:

- *Standard*: We use the official BREAK dataset.

- *Sample Complexity (SC)*: We examine the performance of models with decreasing amounts of training data. The goal is to test which model has better sample complexity.

- *Domain Generalization (DomGen)*: We train on 7 out of 8 sub-domains in BREAK and test on the remaining domain, for each target domain. The goal is to test which model generalizes better to new domains.

As an evaluation metric, we use LF-EM and also the official BREAK metric, NormEM, when reporting test results on BREAK.

### 6.2 Results

**Standard Setup**  Table 2 compares the performance of the different models (§5) to each other and to the top entries on the BREAK leaderboard.

To assess the potential success of the LATENT-RAT architecture, we add an oracle setup (termed LATENT-RAT$_{oracle}$) where learned representations of the **gold** dependencies are fed into the RAT layers. Its outstanding performance (0.759 on the development set), indicates the benefit the sequence-to-sequence model produces from encoding the LF-based dependencies into the tokens representation.

As expected, initializing COPYNET with BERT dramatically improves test performance (0.388→0.47). The LATENT-RAT seq2seq model achieves similar performance (0.471), and the biaffine graph parser, BIAFFINEGP, is slightly behind with an LF-EM of 0.44. Adding an ILP layer on top of BIAFFINEGP to eliminate constraint violations in the output graph improves performance to 0.454. The graph-head of the LATENT-RAT, termed LATENT-RAT$_{graph}$, gets close performance (0.435) to the biaffine graph parser, indicates that the hybrid architecture learns dependency representations.

While our proposed models do not significantly improve performance in the LF-EM setup, we will see next that they improve domain generalization and sample complexity. Moreover, since BIAFFINEGP is a non-autoregressive model that predicts all output edges simultaneously, it dramatically reduces inference time.

| Model | NormEM | | LF-EM | |
|---|---|---|---|---|
| | dev | test | dev | test |
| CopyNet | - | 0.294 | - | 0.388 |
| $BART_{leaderboard\ \#1}$ | - | 0.389 | - | 0.496 |
| CopyNet+BERT | 0.373 | 0.375 | 0.474 | 0.47 |
| BiaffineGP | - | - | 0.441 | 0.44 |
| $BiaffineGP_{ILP}$ | - | - | 0.453 | 0.454 |
| Latent-RAT | 0.356 | 0.363 | 0.469 | 0.471 |
| $Latent-RAT_{graph}$ | - | - | 0.431 | 0.435 |
| $Latent-RAT_{oracle}$ | 0.647 | - | 0.759 | - |

Table 2: Normalized EM and LF-EM on the development and test sets of BREAK.

| Model | ATIS | CLEVR | COMQA | CWQ | DROP | GEO | NLVR2 | SPIDER |
|---|---|---|---|---|---|---|---|---|
| CopyNet+BERT | 0.58 | **0.564** | 0.562 | **0.36** | 0.473 | **0.66** | 0.344 | 0.369 |
| BiaffineGP | **0.591** | 0.489 | 0.595 | 0.322 | 0.445 | 0.62 | 0.293 | **0.41** |
| Latent-RAT | 0.589 | 0.524 | **0.598** | 0.316 | **0.479** | 0.64 | **0.353** | 0.376 |
| CopyNet+BERT | 0.282 | 0.351 | 0.423 | 0.173 | 0.131 | 0.52 | 0.039 | 0.189 |
| BiaffineGP | 0.302 | 0.339 | **0.483** | 0.168 | 0.146 | 0.52 | 0.04 | 0.197 |
| Latent-RAT | **0.335** | **0.356** | 0.435 | **0.189** | **0.149** | **0.58** | **0.063** | **0.201** |
| CopyNet+BERT | -51.38% | -37.77% | -24.73% | -51.94% | -72.30% | -21.21% | -88.66% | -48.78% |
| BiaffineGP | -48.90% | **-30.67%** | **-18.82%** | -47.83% | **-67.19%** | -16.13% | -86.35% | -51.95% |
| Latent-RAT | **-43.12%** | -32.06% | -27.26% | **-40.19%** | -68.89% | **-9.38%** | **-82.15%** | **-46.54%** |

Table 3: Domain Generalization. LF-EM on the development set per sub-domain when training on the entire training set (top), and when training on all domains except the target one (middle). The bottom section is the performance drop from the full setup to the DomGen setup.

Last, the top entry on the BREAK leaderboard uses BART [Lewis et al., 2020], a pretrained seq2seq model (we use a pre-trained encoder only), which leads to a state-of-the-art LF-EM of 0.496.

**Domain Generalization** Table 3 shows LF-EM on each of BREAK's sub-domains when training on the entire dataset (top), when training on all domains but the target domain (middle), and the relative drop compared to the standard setup (bottom). The performance of BiaffineGP and Latent-RAT is higher compared to CopyNet+BERT in the *DomGen* setup. In particular, the performance of Latent-RAT is the best in 7 out of 8 sub-domains, and the performance of BiaffineGP is the best in the last domain. Moreover, Latent-RAT outperforms CopyNet+BERT in all sub-domains. We also observe that the performance drop is lower for BiaffineGP and Latent-RAT compared to CopyNet+BERT. Overall, this shows that using graphs as a source of supervision leads to better domain generalization.

**Sample Complexity** Table 4 shows model performance as a function of the size of the training data. While the LF-EM of BiaffineGP is lower given the full training set (Table 2), when the size of the training data is small it substantially outperforms other models, improving performance by 3-4 LF-EM points given 1%-10% of the data. With 20%-50% of the data Latent-RAT and CopyNet+BERT have comparable performance.

**Inference Time** The graph parser, BiaffineGP, is a non-autoregressive model that predicts all output edges simultaneously, as opposed to a seq2seq model that decodes a single token at each step. We measure the average runtime of the forward pass for both

| Model | 1% | 5% | 10% | 20% | 50% |
|---|---|---|---|---|---|
| CopyNet$_{BERT}$ | 0.112 | 0.261 | 0.323 | 0.38 | 0.426 |
| BiaffineGP | **0.159** | **0.296** | **0.351** | 0.382 | 0.411 |
| Latent-RAT | 0.003 | 0.227 | 0.326 | **0.383** | **0.432** |

Table 4: Development set LF-EM as a function of the size of the training set.

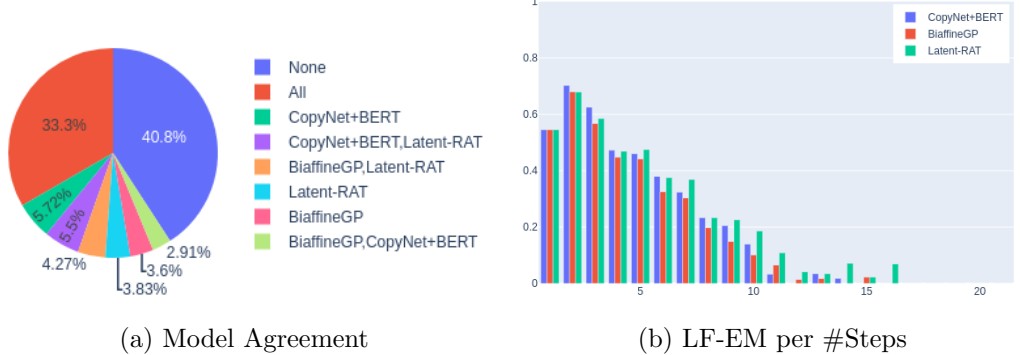

(a) Model Agreement          (b) LF-EM per #Steps

Figure 6: (a) Model agreement in terms of LF-EM on the development set. Each slice gives the fraction of examples predicted correctly by a subset of models. (b) LF-EM on the development set per number of steps. We compute for each example its (gold) number of steps, and calculate the average LF-EM per bin.

BiaffineGP and CopyNet+BERT and find that BiaffineGP has an average runtime of 0.08 seconds, compared to 1.306 seconds of CopyNet+BERT – a 16x speed-up.

### 6.3 Analysis

**Model Agreement**   Figure 6a shows model agreement between the models from §5. Roughly 60% of the examples are predicted correctly by one of the models, indicating that ensemble of the three models could result in further performance improvement. The agreement of Latent-RAT with CopyNet+BERT (5.5%) and BiaffineGP (4.27%) is greater than the agreement of CopyNet+BERT with BiaffineGP, perhaps since it is a hybrid of a seq2seq and graph parser.

**Length Analysis**   We compared the average LF-EM of models for each possible number of steps in the QDMR structure (Fig. 6b). We observe that CopyNet+BERT outperforms Latent-RAT when the number of steps is small, but once the number of steps is $\geq 5$, Latent-RAT outperforms CopyNet+BERT, showing it handles complex decompositions better, and in agreement with the tendency of seq2seq models to struggle with long output sequences.

**Error Analysis**   We manually analyzed randomly sampled errors from each model (§A.8). For all models, the largest error category (34-72%) is actually cases where the prediction is correct but not captured by the LF-EM metric, showing that the performance of current QDMR parsers is actually quite high.

## 7. Conclusion

In this work, we propose to represent QDMR structures with a dependency graph over the input tokens, and propose a graph parser and a seq2seq model that uses graph supervision as an auxiliary loss. We show that a graph parser is 16x faster than a seq2seq model, and that it exhibits better sample complexity. Moreover, using graphs as auxiliary supervision improves out-of-domain generalization and leads to better performance on questions that represent a long sequence of computational steps. Last, we propose a new evaluation metric for QDMR parsing and show it better corresponds to human intuitions.

## Acknowledgments

We thank Vivek Kumar Singh for his helpful ILP guidelines, and Tomer Wolfson for having kindly assisted in running our evaluation metric on our predictions for BREAK test set. This research was partially supported by The Yandex Initiative for Machine Learning, and the European Research Council (ERC) under the European Union Horizons 2020 research and innovation programme (grant ERC DELPHI 802800).

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

## Appendix A. Appendices

### A.1 QDMR LF

Table 5 shows the different operators, their properties and examples of LFs. Table 6 shows terms that are used to identify the QDMR step operator's properties. We use the same lexicon from BREAK [Wolfson et al., 2020] for detecting operators, extended with some specifications for numeric properties such as *equals_0*.

### A.2 LF-Based Evaluation (LF-EM)

In §3 we describe a LF-based evaluation metric that is based on normalization steps. We now elaborate the normalization process.

Given a logical form $Z$, we transform each step to a normalized form, and the final textual representation is given by representing each step as described in §3: OPERATOR[*property*](arg=...; ...). We apply the following steps (Fig. 7):

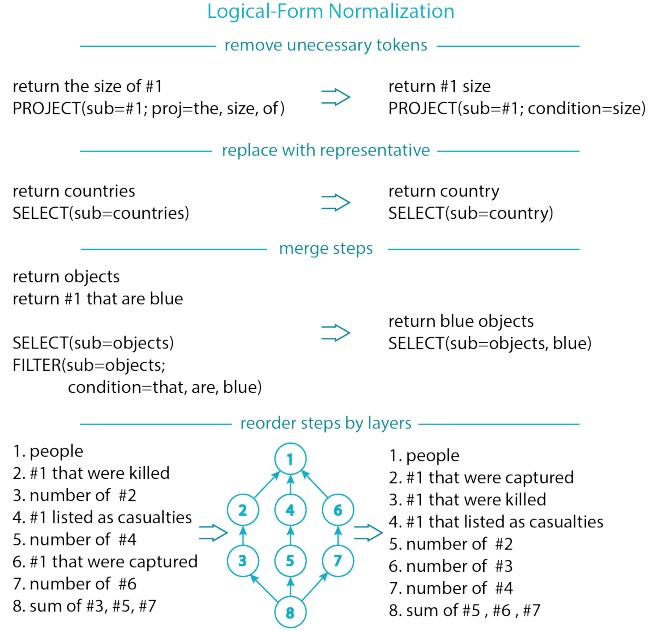

Figure 7: An illustration of LF normalization. Normalization is done on the LF $Z$, and we present QDMR steps for ease of reading.

**Remove and normalize tokens**  Each LF step includes a list of tokens in its arguments. In this normalization step, we remove lexical items, such as *"max"*, which are used to detect the operator and property (Table 6 in §A.1), as those are already represented outside the arguments. In addition, we remove words from a stop word list ($\mathcal{V}_{aux}$, see Fig. 3). Finally, we use a synonym list to represent words in such a list with a single representative (*countries→country*, see Table 7).

| Operator | PROP | ARG | Example |
|---|---|---|---|
| SELECT | ∅ | sub | return cubes
SELECT[](sub=cubes) |
| FILTER | ∅ | sub, condition | return #1 from Toronto
FILTER[](sub=#1, cond=from Toronto) |
| PROJECT | ∅ | sub, projection | return the head coach of #1
PROJECT[](sub=#1, projection=the head coach of) |
| AGGREGATE | *max, min, count, sum, avg* | arg | return maximal number of #1
AGGREGATE[*max*](arg=#1) |
| GROUP | *max, min, count, sum, avg* | key, value | return the number of #2 for each #1
GROUP[*count*](key=#1, value=#2) |
| SUPERLATIVE | *max, min* | sub, attribute | return #2 where #3 is the lowest
SUPERLATIVE[*min*](sub=#2, attribute=#3) |
| COMPARATIVE | *equals, equals-[0/1/2], more, more-than-[0/1/2], less, less-than-[0/1/2]* | sub, attribute, condition | return #1 where #2 is more than 100
COMPARATIVE[*more*](sub=#1, attribute=#2, condition=100) |
| COMPARISON | *max, min, count, sum, avg, true, false* | arg | return which is higher of #1, #2
COMPARISON[*max*](arg=#1, arg=#2) |
| UNION | ∅ | sub | return #1, #2
UNION[](sub=#1, sub=#2) |
| INTERSECTION | ∅ | intersect, projection | return parties in both #2 and #3
INTERSECTION[](intersect=#2, intersect=#3, projection=parties) |
| DISCARD | ∅ | sub, exclude | return #1 besides #2
DISCARD[](sub=#1, exclude=#2) |
| SORT | ∅ | sub, order | return #1 ordered by name
SORT[](sub=#2, order=name) |
| BOOLEAN | *equals, equals-[0/1/2], more-than-[0/1/2], less-than-[0/1/2], and-true, and-false, or-true, or-false, if-exists* | sub, condition | return if #1 is the same as #2
BOOLEAN[*equals*](sub=#1, condition=#2) |
| ARITHMETIC | *sum, diff, multiply, div* | arg, left, right | return the difference of #3 and #4
ARITHMETIC[*diff*](left=#3, right=#4) |

Table 5: LF operators, properties and arguments. Each QDMR step can be mapped to one of the above operators, where its LF consists of its operator, properties and arguments. The example column shows an example for such LF.

| Operator | PROP | Lexical entries |
|---|---|---|
| AGGREGATE, COMPARISON, GROUP | *max* | max, most, more, last, bigger, biggest, larger, largest, higher, highest, longer, longest |
| AGGREGATE, COMPARISON, GROUP | *min* | min, least, less, first, fewer, smaller, smallest, lower, lowest, shortest, shorter, earlier |
| AGGREGATE, COMPARISON, GROUP | *count* | count, number of, total number of |
| AGGREGATE, ARITHMETIC, COMPARISON, GROUP | *sum* | sum, total |
| AGGREGATE, COMPARISON, GROUP | *avg* | avg, average, mean |
| ARITHMETIC | *diff* | difference, decline |
| ARITHMETIC | *multiply* | multiplication, multiply |
| ARITHMETIC | *div* | division, divide |
| BOOLEAN, COMPARATIVE | *equals* | equal, equals, same as |
| BOOLEAN | *if-exists* | any, there |
| COMPARATIVE | *more* | more, at least, higher than, larger than, bigger than |
| COMPARATIVE | *less* | less, at most, smaller than, lower than |
| SUPERLATIVE | *max* | most, biggest, largest, highest, longest |
| SUPERLATIVE | *min* | least, fewest, smallest, lowest, shortest, earliest |

Table 6: Property lexicon. Tokens for detecting the properties of a QDMR step, for creating its logical form.

**Merge Steps** QDMR annotations sometime vary in their granularity. For example one example might contain *"return metal objects"*, while another might have *"return objects; return #1 that are metal"*. This is especially common in FILTER and PROJECT steps. We merge chains of FILTER steps, as well as FILTER or PROJECT steps that follow a SELECT step.
.

**Reorder steps** QDMR describes a directed acyclic graph of computation steps, and there are multiply ways to order the steps (Fig. 7). We recursively compute the *layer* of each step as $\text{layer}(s) = \max_{s \to s'} \{\text{layer}(s')\} + 1$, where the maximization is over all the steps $s$ refers to. We then re-order steps by layer and then lexicographically.

We manually evaluate the metrics NormEM and LF-EM on 50 random development set examples using predictions from the CopyNet-BERT model (see §6). We find that both (binary) metrics have perfect precision: they only assign credit when indeed the QDMR reflects the correct question decomposition, as judged by the authors. However, LF-EM covers more examples, where the LF-EM on this sample is 52.0, while NormEM is 40.0. Thus, LF-EM provides a tighter lower bound on the performance of a QDMR parser and is a better metric for QDMR parsing.

| Type | Equivalence Class |
|---|---|
| Modifications | cube, cubes, ... |
| | old, oldness, ... |
| | taller, tall, ... |
| | working, work, ... |
| Operational | biggest, longest, highest, ... |
| Synonyms | elevation, height |
| | 0, zero |
| | ... |

Table 7: BREAK Equivalence Classes. (1) *Modifications* - the same modifications of the question tokens that were used for creating BREAK annotation lexicon (e.g plural/singular form, nounify adjectives, lemmatize adjectives, lemmatize verbs); (2) *Operational* equivalence induced from properties lexicon; (3) Manually-defined *Synonyms* lexicon; We mostly retrieve the final equivalence classes by merging classes that share some tokens.

### A.3 Token Alignment

We denote the question tokens by $q = q_1 \ldots q_n$ and the $i$th QDMR step tokens by $\forall i \in [1..m], s^i = s_1^i \ldots s_{n_i}^i$. An *alignment* is defined by $M = \{(q_i, s_j^k) \mid q_i \approx s_j^k; i \in [1..n], k \in [1..m], j \in [1..n_k]\}$, where by $t \approx t'$ we mean $t, t'$ are either identical or equivalent. Roughly speaking, these equivalences are based on the BREAK annotation lexicon (Fig. 3) – in particular, the inflections of the question tokens $\mathcal{V}_q$ (e.g , *"object"* and *"objects"*), and equivalence classes on top of the constant lexicon $\mathcal{V}_{const}$ (e.g , *"biggest"* and *"longest"*). See Table 7 (§A.2) for more details.

To find the best alignment $M$, we formulate an optimization problem in the form of an Integer Linear Program (ILP) and use a standard ILP solver.[2] The full details are given as part of our open source implementation. The objective function uses several heuristics to assign a high score to an alignment that has the following properties (Fig. 8):

- *Minimalism*: Aligning each question token to at most one QDMR step token and vice versa is preferable.
- *Exact Match*: Aligning a question token to a QDMR token that is identical is preferable.
- *Sequential Preference*: Aligning long sequences from the question to a single step is preferable. When a step has a reference token (*#1*), we take into account the tokens in the referenced step (see Fig. 8, top right).
- *Steps Coverage*: Covering more steps is preferable.

### A.4 Spans Dependencies Extraction

Given the QDMR, LF, and alignment $M$, we construct the Span Dependency Graph (SDG). Each QDMR step is a node labeled by a list of tokens (spans). The list of tokens is computed with the alignment $M$, where given a QDMR step $s^k$, the list contains all question tokens

---

2. https://developers.google.com/optimization

If a body of water is visible in the right image of a water buffalo.

1. return water buffalo
2. return the right image of #1
3. return body of water
4. return if #3 is visible in #2

(a) Sequential Preference

Show the school name and driver name for all school buses.

1. return school buses
2. return schools of #1
3. return names of #2
4. return drivers of #1
5. return names of #4
6. return #3, #5

(b) Steps Coverage

Give the country id and corresponding count of cities in each country.

1. return countries
2. return country ids of #1
3. return cities of #1
4. return number of #3 for each #1
5. return #2, #4

If all of the gorillas are holding leaves in their left hand.

1. return gorillas
2. return leaves
3. return hand of #1
4. return #3 that is left
5. return #1 holding #2 in #4
6. return the number of #1
7. return the number of #5
8. return if #6 is equal to #7

(c) Exact Match

Figure 8: Heuristics for token alignment. Potential tokens for alignment colored, where the preferable choice according to the heuristic is underlined. On the top left, the second occurrence of *"water"* is preferred in QDMR step #1 due to the adjacent word *"buffalo"*. On the top right, the second occurrence of *"name"* is preferred in QDMR step #5, because this step refers to #4 that contains the word *"drivers"*.

$q_i$, such that $(q_i, s_j^k) \in M$, where $s_j^k$ is a word in $s^k$. The list is ordered according to the position in the question.

Edges in the SDG are computed using reference tokens. If step $s^i$ has a reference token to step $s^j$, we add a directed edge $(s^i, s^j)$ (we abuse notation and refer to SDG nodes and QDMR steps with the same notation). Each edge has a *label*, which is a triple consisting of the operator $o^i$ of the source node $s^i$, the property $\rho^i$ of the source node, and the named argument $\eta_{\text{ref}}^i$ that contains the reference token. For readability we denote the label triplet $label(i, j) = \langle o^i, \rho^i, \eta_{\text{ref}}^i \rangle$ by $o^i\text{-}\eta_{\text{ref}}^i[\rho^i]$. Figure 4 shows an extracted SDG.

## A.5 SDG→DG

We construct a DG by projecting the SDG on the question tokens. This is done by: (a) For each SDG node and its list of tokens, add edges between the tokens from left-to-right with a new `span` label (black edges in Fig 4); (b) use the rightmost word in every span as its representative for the edges between different spans.

However, this transformation is non-trivial for two reasons. First, some SDG nodes do not align to any question token. Second, some question tokens align to multiple SDG nodes, which does not allow the DG to be converted back to an SDG unambiguously for evaluation. We resolve such representation issues by adding special tokens at the end of the sequence and using them as extra tokens for alignment. We give the details in §A.6.

## A.6 DG Representation Issues

In §A.5 we describe the conversion procedure from SDG to DG. This transformation is non-trivial for two reasons. First, some SDG nodes do not align to any question token. Second, some question tokens align to multiple SDG nodes, which does not allow the DG to be converted back to an SDG unambiguously for evaluation.

We now explain how we resolve such representation issues, mostly based on adding more tokens to the input question. Fig. 9 illustrates the different types of challenges and our proposed solution.

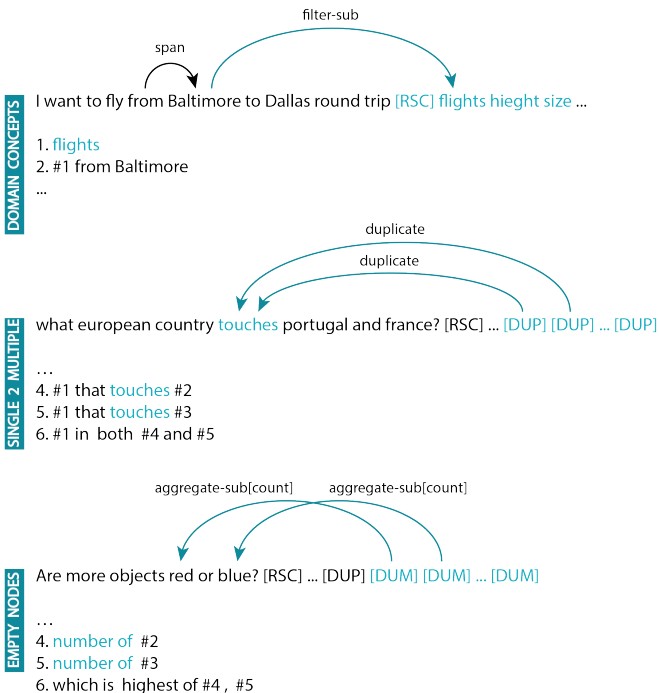

Figure 9: Representation Issues. Projecting the SDG over the question token (DG) is not always trivial. We solve this by concatenating special tokens to the question.

**Domain-specific concepts**  QDMR annotators were allowed to use a small number of tokens that are pragmatically assumed to exist in the domain ($\mathcal{V}_{store}$ in Fig. 3). For example, when annotating ATIS questions [Hemphill et al., 1990], the word *"flight"* is allowed to be used in the QDMR structure even if it does not appear in the question, since this is a flight-reservation domain. We concatenate all the words in $\mathcal{V}_{store}$ to the end of each question after a special separator token, which allows token alignment (§A.3) to map such QDMR steps to a question word (Fig. 9, top).

**Empty SDG nodes**  Some steps only contains tokens that are not in the question (e.g., *"Number of #2"* in Fig. 9 bottom), and thus their list of tokens in the SDG node is empty. In this case, we cannot ground the SDG node in the question. Therefore we add a constant number of *dummy tokens*, *[DUM]*, which are used to ground such SDG nodes.

**Single tokens to multiple SDG nodes**  A single question token can be aligned to multiple SDG nodes. Recall the tokens of each SDG nodes are connected with a chain of `span` edges. This leads to cases where two chains that pass through the same question token cannot be distinguished when converting the DG back to an SDG for evaluation. We solve this by concatenating a constant number of special *[DUP]* tokens that conceptually duplicate another token by referring to it with a new `duplicate` label. Now, each span chain uses a different copy of the shared token by referring to the *[DUP]* instead of the original one.

### A.7 Experiments Parameters

**CopyNet+BERT**   The LSTM decoder has hidden size 768. We use a batch size of 32 and train for up to 25 epochs (∼35k steps) with beam search of size 5.

**BiaffineGP**   The POS embeddings are of size 100. The four FFNs consist of 3-layers with hidden size 300 and use ELU activation function. We use dropout of rate 0.6 on the contextualized encodings, and of rate 0.3 on the FF representations. We use a batch size of 32 and train for up to 80 epochs (∼111k steps).

**Latent-RAT**   We stack 4 relation-aware self-attention layers on top of the contextualized encodings, each with 8 heads and dropout with rate 0.1. The FFNs for relation representation uses 3-layers with hidden size of 96, ReLU activation function and dropout rate of 0.1. We tie the layers, and multiply the graph loss by 100. The rest is identical to the CopyNet-BERT configuration.

**Optimization**   We used the Adam optimizer [Kingma and Ba, 2015] with the default hyperparameters. The learning rate changes during training according to the slanted triangular schema, in which it linearly increases from 0 to $lr$ for the first $warmup\_steps = 0.06 \cdot max\_steps$, and afterwards linearly decreases back to 0. We use learning rate of $1 \cdot 10^{-3}$, and a separate learning rate of $5 \cdot 10^{-5}$ for the BERT-based encoder.

### A.8 Error Analysis

We randomly sampled 50 errors from each model and manually analyzed them. Table 8 describes the error classes for each model, and Appendix §A.9 provides examples for these classes. Each example can have more than one error category.

For all models, the largest error category is actually cases where the prediction is correct but not captured by the LF-EM metric: 56% for CopyNet+BERT, 34% for BiaffineGP and 72% for Latent-RAT. This shows that the performance of current QDMR parsers is actually quite high, but capturing this with an automatic evaluation is challenging.

When taking a more loose definition of correctness (termed *"Correct (soft)"*), the rates increase to: 62% for CopyNet+BERT, 50% for BiaffineGP and 80% for Latent-RAT. In these cases we focus on the final returned result correctness, and ignore unused steps, duplicate steps or references, implicit information (commonsense), etc. §A.10 provides some examples for such predictions.

Cases where the output is correct include:

- *Equivalent Solutions*: the prediction is logically equivalent to the gold structure.

- *Elaboration Level*: the model prediction is more/less granular compared to the gold structure, but the prediction is correct.

- *Redundancy*: additional information is predicted/omitted that does not effect the computation. For example the second occurrence of *"yards"* in *"2. return **yards** of #1; 3. #1 where #2 is lower than 10 **yards**"*.

- *Wrong Gold* - cases where the predication is more accurate than the gold decomposition. The main classes of errors are:

- *Missing Information*: missing steps, missing references or missing tokens that affect the result of the computation.

|                         | CopN    | BiGP    | L-RAT   |
|-------------------------|---------|---------|---------|
| Correct                 | 56.00%  | 34.00%  | 72.00%  |
| Correct (soft)          | 62.00%  | 50.00%  | 80.00%  |
| Equivalent Solutions    | 34.00%  | 20.00%  | 44.00%  |
| Elaboration Level       | 22.00%  | 16.00%  | 26.00%  |
| Redundancy              | 8.00%   | 2.00%   | 2.00%   |
| Wrong Gold              | 2.00%   | 6.00%   | 6.00%   |
| Missing Information     | 18.00%  | 24.00%  | 10.00%  |
| Additional Steps        | 12.00%  | 16.00%  | 6.00%   |
| Wrong Global Structure  | 10.00%  | 30.00%  | 12.00%  |
| Wrong Step Structure    | 0.00%   | 18.00%  | 4.00%   |
| Out of Vocabulary       | 10.00%  | 0.00%   | 4.00%   |

Table 8: Error classes and their frequency over a sample of 50 random errors. Model names were shortened from CopyNet+BERT, BiaffineGP and Latent-RAT.

- *Additional Steps*: duplicate steps or additional steps that change the result of the computation.

- *Wrong Global Structure*: The computation described by the predicted structure is wrong (for example, addition instead of subtraction).

- *Wrong Step Structure*: incoherent structure of a particular step that cannot be mapped to a proper structure.

- *Out of Vocabulary*: seq2seq models sometimes predict tokens that are not related to the question nor the decomposition. For example, "rodents" in a question about flowers.

The seq2seq models preserve better global and local structure (*Wrong Global Structure, Wrong Step Structure*). The graph parser, by design, has no *Out of Vocabulary* tokens and less *Redundancy*, but suffers from incoherence (*Additional Steps, Wrong Global Structure, Wrong Step Structure*) due to non-autoregressiveness. The combined architecture, Latent-RAT seems to utilize the dependence information to improve *Redundancy* and *Out of Vocabulary* issues as well as the additional/missing information (*Missing Information, Additional Steps*) compared to the seq2seq model it is based on - CopynNet+BERT.

## A.9 Error Analysis Examples

Some examples for each error class from §A.8. The gold decompositions are given on left, and the predictions are on the right.

### Equivalent Solution

How many yards longer was the longest field goal over the second longest?

| | |
|---|---|
| 1. select(sub=field goals)
2. project(projection=yards of #REF; sub=#1)
3. aggregate[max](arg=longest #2)
4. aggregate[max](arg=second longest #2)
5. arithmetic[difference](left=#3; right=#4) | 1. select(sub=field goals)
2. project(projection=yards of #REF; sub=#1)
3. aggregate[max](arg=#2)
4. discard(exclude=#3; sub=#2)
5. aggregate[max](arg=#4)
6. arithmetic[difference](left=#3; right=#5)    CopyNet |

If there are exactly two fluffy dogs and no reflections.

| | |
|---|---|
| 1. select(sub=dogs)
2. filter(condition=that are fluffy; sub=#1)
3. aggregate[count](arg=#2)
4. boolean[equals_2](condition=is equal to two; sub=#3)
5. select(sub=reflections)
6. aggregate[count](arg=#5)
7. boolean[equals_0](condition=is equal to zero; sub=#6)
8. boolean[logical_and,true](sub=#4,#7) | 1. select(sub=dogs)
2. filter(condition=that are fluffy; sub=#1)
3. aggregate[count](arg=#2)
4. boolean[equals_2](condition=is equal to two; sub=#3)
5. project(projection=reflections of #REF; sub=#2)
6. aggregate[count](arg=#5)
7. boolean[equals_0](condition=is equal to zero; sub=#6)
8. boolean[logical_and,true](sub=#4,#7)    Latent-RAT |

### Elaboration Level

What tv program with more than 19 episodes did Joey Lawrence play on?

| | |
|---|---|
| 1. select(sub=Joey Lawrence)
2. project(projection=tv programs of #REF; sub=#1)
3. filter(condition=with more than 19 episodes; sub=#2) | 1. select(sub=Joey Lawrence)
2. project(projection=tv program; sub=#1)
3. project(projection=episodes; sub=#2)
4. group[count](key=#2; value=#3)
5. comparative[more](attribute=#4; condition=more than 19; sub=#2)    BiaffineGP |

### Redundancy

How many TD passes were under 10 yards?

| | |
|---|---|
| 1. select(sub=TD passes)
2. project(projection=yards of #REF; sub=#1)
3. comparative[less](attribute=#2; condition=is lower than 10 yards; sub=#1)
4. aggregate[count](arg=#3) | 1. select(sub=TD passes)
2. project(projection=yards of #REF; sub=#1)
3. comparative[less](attribute=#2; condition=is lower than 10; sub=#1)
4. aggregate[count](arg=#3)    CopyNet |

### Wrong Gold

How many objects are either yellow or shiny?

1. select(sub=objects)
2. filter(condition=that are yellow; sub=#1)
3. filter(condition=that are shiny; sub=#1)
4. aggregate[count](arg=#2)
5. aggregate[count](arg=#3)
6. arithmetic[sum](arg=#4,#5)

1. select(sub=objects)
2. filter(condition=shiny; sub=#1)
3. discard(exclude=#2; sub=#1)
4. aggregate[count](arg=#2)
5. filter(condition=yellow; sub=#3)
6. aggregate[count](arg=#5)
7. arithmetic[sum](arg=#4,#6)

BiaffineGP

### Missing Information

What shape of the only object that wont roll if pushed?

1. select(sub=objects)
2. filter(condition=that wont roll if pushed; sub=#1)
3. project(projection=shape of #REF; sub=#2)

1. select(sub=objects)
2. filter(condition=that has roll if pushed; sub=#1)
3. project(projection=shape of #REF; sub=#2)

CopyNet

### Additional Steps

What is the smallestt shape and also yellow?

1. select(sub=shapes)
2. project(projection=size of #REF; sub=#1)
3. superlative[min](attribute=#2; sub=#1)
4. filter(condition=that are yellow; sub=#3)

1. select(sub=shape)
2. comparative(condition=smallestt; sub=#1)
3. project(projection=size; sub=#1)
4. superlative[min](attribute=#3; sub=#1)
5. filter(condition=yellow; sub=#2,#4)

BiaffineGP

If at least five orange dogs without collars sit upright in a row, gazing intently, in one image, and the other image includes dogs in collars arranged more or less in a row.

1. select(sub=one image)
2. project(projection=dogs in #REF; sub=#1)
3. filter(condition=that are orange; sub=#2)
4. select(sub=collars)
5. filter(condition=#4,without; sub=#3)
6. filter(condition=that sit upright; sub=#5)
7. filter(condition=in a row; sub=#6)
8. filter(condition=that are gazing intently; sub=#7)
9. aggregate[count](arg=#8)
10. boolean(condition=is at least five; sub=#9)
11. select(sub=the other image)
12. project(projection=dogs in #REF; sub=#11)
13. filter(condition=#4,in; sub=#12)
14. boolean(condition=are arranged more or less in a row; sub=#13)
15. boolean[logical_and,true](sub=#10,#14)

1. select(sub=one image)
2. project(projection=dogs in #REF; sub=#1)
3. filter(condition=that are orange; sub=#2)
4. select(sub=collars)
5. filter(condition=that are orange; sub=#3)
6. filter(condition=that are in a row; sub=#5)
7. filter(condition=that are gazing intently; sub=#6)
8. aggregate[count](arg=#7)
9. boolean(condition=is at least five; sub=#8)
10. select(sub=other image)
11. project(projection=dogs in #REF; sub=#10)
12. project(projection=collars of #REF; sub=#10)
13. filter(condition=that are arranged more in a row; sub=#12)
14. aggregate[count](arg=#13)
15. boolean(condition=is at least five; sub=#14)
16. boolean[logical_and,true](sub=#8,#15)

Latent-RAT

### Wrong Global Structure

How many was the difference beween Sobieski's force and the Turks and Tatars?

1. select(sub=Sobieski)
2. project(projection=the force of #REF; sub=#1)
3. project(projection=size of #REF; sub=#2)
4. select(sub=the Turks and Tatars)
5. project(projection=the force of #REF; sub=#4)
6. project(projection=size of #REF; sub=#5)
7. arithmetic[difference](left=#6; right=#3)

1. select(sub=Sobieski)
2. project(projection=force of #REF; sub=#1)
3. select(sub=Turks)
4. select(sub=the Tatars)
5. arithmetic[difference](left=#2; right=#3)
6. arithmetic[difference](left=#4; right=#5)

Latent-RAT

### Wrong Step Structure

How many years after Knopf was founded was it officiaully incorporated?

1. select(sub=Knopf was founded)
2. select(sub=Knopf was officiaully incorporated)
3. project(projection=year of #REF; sub=#1)
4. project(projection=year of #REF; sub=#2)
5. arithmetic[difference](left=#4; right=#3)

1. project(projection=Knopf was founded years; sub=#1)
2. select(sub=was it officiaully incorporated)
3. project(projection=years; sub=#2)
4. arithmetic[difference](left=#3; right=#1)

BiaffineGP

### Out of Vocabulary

What coutry is currently led by an acting prime minister and is a part of NATO?

1. select(sub=NATO)
2. project(projection=coutries of #REF; sub=#1)
3. filter(condition=that are currently led by an acting prime minister; sub=#2)

1. select(sub=company)
2. filter(condition=that is currently led by an acting prime minister;  sub=#1)
3. filter(condition=that is part of NATO; sub=#2)

CopyNet

## A.10 Correct (Soft) Examples

Examples for semi-correct predictions (§A.8). For convenience, some of the LFs are given in a QDMR form.

Commonsense

when did alex rodriguez join the seattle marnors?

1. return alex rodriguez plays the seattle marnors
2. return date of #1                                                    CopyNet

If the left image shows a small brown pony hitched to a cart without a rider in it.

1. sreturn pony
2. return #1 that is small
3. return #2 that is brown
4. return cart
5. return #3 hitched to #4
6. return rider
7. return #5 without #6 in it
8. return if #7 is in the left image                                   CopyNet

How many was the difference beween Sobieski's force and the Turks and Tatars?

1. return Sobieski
2. return force of #1
3. return Turks
4. return the Tatars
5. return the difference of #2 and #3
6. return the difference of #4 and #5                                  Latent-RAT

Implicit Information

How many points did the 49ers win by in the NFC championship against the Chicago Bears?

1. return the 49ers
2. return points that #1 win by
3. return #2 in the NFC championship against the Chicago Bears
(missing: return number of #3)                                         CopyNet

what was julius caesar's mums name?

1. return julius caesar
2. return mum of #1
(missing: return name of #2)                                           Latent-RAT

### Bad Global Structure

Show name, country, age for all singers ordered by age from the oldest to the youngest.

1. return singers
2. return name of #1
3. return country of #1
4. return age of #1
5. return age of #1
6. return #2, #3, #4, #5
7. return #6 ordered by #4,#5 from oldest to youngest            BiaffineGP

What is the smallestt shape and also yellow?

1. select(sub=shape)
2. comparative(condition=smallestt; sub=#1)
3. project(projection=size; sub=#1)
4. superlative[min](attribute=#3; sub=#1)
5. filter(condition=yellow; sub=#2,#4)            BiaffineGP

If none of the sneakers in the images are on a foot.

1. return sneakers in the images
2. return foot
3. return #1 on #2
4. return if #3 are
5. return number of #3
6. return if #5 equals 0            BiaffineGP

### Bad Step Structure

What color is the object that is to the right of the gray cylinder and in front of the red sphere?

1. select(sub=the gray cylinder)
2. select(sub=the red sphere)
3. filter(condition=#1,object that is to the right of; sub=#3)
4. filter(condition=#2,in front of; sub=#3)
5. project(projection=color; sub=#4)            BiaffineGP

If there are snow dog sleds with riders on board them.

1. return snow dog sleds
2. return riders
3. return #2 that are on board of #1,#1
4. return number of #3
5. return if #4 is at least two            Latent-RAT

