# OpenReview forum: "Question Decomposition with Dependency Graphs"
_AKBC.ws/2021/Conference — AKBC 2021_

### Official Review · Reviewer_Y2Ua · 2021-07-21
**Thorough work on dependency-based QDMR parsing**

**Rating:** 8
**Confidence:** 3

**Review:**

This paper is in the area of Question Decomposition Meaning Representation (QDMR), originally introduced by Wolfson et al (2020). The core idea proposed in this paper is a new representation based on dependency graphs; based on such a graph, QDMR logical forms can be computed using a deterministic procedure. In addition, the paper describes an evaluation protocol based on the logical forms. Finally, the paper also introduces two different automatic parsing methods that produce the dependency graphs, one of which uses the well-known biaffine approach by Dozat et al., and one using an autoregressive approach based on the Relation-Aware Transformer; the biaffine parser is faster but somewhat less accurate.

This is solid work that covers quite a bit of ground. It seems well-justified to pose the QDMR prediction task using a graph-based representation, and the evaluation protocol based on logical forms is well justified; the conversion algorithms are described thoroughly in the supplemental material.

---

> ### Author Response · Authors · 2021-07-30
> **Response to Reviewer Y2Ua**
>
> Thank you for finding our new representation is reasonable, and for your interest in our work. We appreciate your review!

---

### Official Review · Reviewer_h9MV · 2021-07-22
**Solid approach to QDMR parsing using dependency graphs**

**Rating:** 7
**Confidence:** 3

**Review:**

The authors propose a novel view of decomposing questions into QDMR steps using dependency parsing. Specifically, they propose representing QDMR decompositions as a dependency graph in which the edges correspond to computation steps. Using this representation, they present two new methods for decomposing questions; a biaffine graph parser that directly predicts all of the edges, and a seq2seq parser using a relation-aware transformer to incorporate dependency information as additional supervision. The first model shows a significant speed-up at the cost of some accuracy, while the second model is shown to improve generalization performance on unseen domains. Additionally, the authors propose a new evaluation metric for QDMR parsing that better correlates with human judgements.

Pros:
- Novel representation for QDMR as a dependency graph
- Two models making use of the above representation to achieve specific improvements in runtime/accuracy
- Good amount of explanation and analysis of the models' performance.

Cons:
- The approaches don't provide a significant improvement on in-domain QDMR parsing when the full dataset is used.
- Lacking analysis of the LF-EM evaluation compared to the prior NormEM. Specifically, I think it would be helpful to see some examples and explanation of the cases that LF-EM covers that a re not covered by NormEM.

Questions:
- Is there a reason you didn't include any ensembled results? Based on model agreement, as you mention, combined with the fact that the Latent-RAT model specifically performs better on longer decompositions, it seems ensembling might be a way to improve over  the SOA.
- Have you run any extrinsic evaluations of these approaches? Given the error analysis that automatic evaluations of QDMR is missing a lot of correct cases, I'd be curious to see how the predictions of each of them affects the end QA performance.

---

> ### Author Response · Authors · 2021-07-30
> **Response to Reviewer h9MV**
>
> Thank you for finding our new representation is novel and for your interest in our proposed evaluation metric. We appreciate your detailed review!
>
> Regarding the ensembled results:
> We tried a light black-box ensemble method, in which a sample is first classified by a BERT-based classifier with a specific model, and then the inference is done by that model. This technique didn’t lead to a significant improvement. We believe a white-box technique, in which the models' confidences are taken into account, will work better here, but decided to leave it for future work.
>
> “extrinsic evaluations of these approaches”
> In this work we focused on comparing the “traditional” seq2seq-based approaches with our new graph-based approach using an automatic metric, just like the official one. We agree that showing the effect on QA performance could be a very positive signal for correctness, but it's not a guarantee for correctness - so we preferred to leave it for future work and validate the metric through human judgement. Moreover, we tried to be agnostic to the downstream use of QDMR, as QA is not the only usage it may have (e.g. as an auxiliary task in a QA system, or for data augmentation through example generation), similar to how meaning representations are typically evaluated (AMR, SRL, QA-SRL) using intrinsic measure.

---

### Official Review · Reviewer_NVb5 · 2021-07-22
**Nice paper which could be better with some more discussion**

**Rating:** 7
**Confidence:** 4

**Review:**

The paper investigates the utility of dependency graph (DG) representation in converting questions to Question Decomposition Meaning Representation (QDMR) annotations. As an intermediate step, the paper proposes to convert QDMR annotations to logical form (LF) representation (this was also done in the original QDMR paper). This LF representation is used to then extract a DG representation from it in a deterministic manner. The DG can also be converted to LF in a similar manner. Additionally, the paper also proposes a LF based evaluation metric which apparently is a better measure for semantic equivalence between QDMRs as compared to the previously proposed metric.

To utilize DGs, the paper proposes two ways. (1) a graph-based parser that predicts all the DG edges simultaneously with some ILP based inference to ensure that the predicted DG is valid. This parser is not the most accurate but extremely fast. (2) Latent-RAT: Use the DG as side-information in a seq2seq transformer model by using RAT transformer layers [Shaw et al., 2018, Wang et al., 2020] to explicitly model relations between tokens. The model directly outputs QDMR annotation.

Experimentally, the paper shows that the Latent-RAT model achieves competitive performance compared to the previous CopyNet+BART model in an in-domain test setting. In a domain adaptation setting (train on 7 domains in QDMR dataset, test on the remaining one), both graph-based parser and Latent-RAT achieve significant improvements compared to the baseline. The performance on each domain is still way behind when in-domain training data is used though. Analysis experiments show that graph-based parser learns better in low-data scenarios, and that the latent-RAT model performs better on longer LFs.

Overall, I think the ideas presented in the paper are reasonable and well tested empirically that it should warrant acceptance. I do not find any concerning issues with the paper to advocate for rejection. Though, I do have various questions that I think the authors should try to address in the paper:
1. What are the issues/challenges in converting QDMR/LF annotations to DG representations? Step 1 of token alignment seems an avenue where it is easy to make mistakes.
2. Are the resulting DGs always projective or do they have crossing edges?
3. Did the authors think about using a Shift-Reduce parser to predict the DGs? Is there any intuition on the expected performance?
4. Similarly, do you think the model will achieve better performance if predicting LFs in the latent-RAT model? It seems that the structured nature of LFs might give better inductive bias while learning.
5. I think the paper is missing an important experiment of using the exact same transformer architecture but not using the additional loss from DGs. This will help tease apart the contribution from the NN architecture and the additional benefit from the DG signal.
6. The second last line of section 4. states “We now describe the details of the LF→DG conversion” but I do not find this anywhere. Maybe I’m missing something?
7. In the main paper and A.2, the paper does not describe the cases in which LF-EM gets higher coverage than NormEM and in which ways it is lacking. Discussion around this will promote better future research by the readers.

---

> ### Author Response · Authors · 2021-07-30
> **Response to Reviewer NVb5**
>
> Thank you for finding our new approach interesting and for your detailed review!
>
> “issues/challenges in converting QDMR/LF annotations to DG”:
> For brevity, we describe the main challenges of this conversion in the Appendices (A3, A6). We split these challenges into two types: (1) Token Alignment ambiguity (A3, see Fig. 8); (2) Graph extraction (A6), especially dealing with implicit information that is missing in the question, and with "shared" tokens of different execution paths - which make these paths indistinguishable from each other (see Fig.9).
>
> “DGs always projective”:
> Our dependency graphs may have crossing edges (see Fig.4 bottom right, the pink edge crosses the red and the orange edges).
>
> “Shift-Reduce parser to predict the DGs”:
> We haven't tried it. We were interested in this work in a non-autoregressive approach to take advantage of its fast inference. We agree that designing a shift-reduce system that will produce QDMR dependency graphs is an interesting direction for future work, and might improve performance and consistency (see Error Analysis, A.8, Table 8, "wrong global/step structure").
>
> “ better performance if predicting LFs “:
> Yes, The seq2seq models, including Latent-RAT, perform better (~2 LF-EM points) when  trained to predict a sequential representation of the LF directly. We didn't include this part in the paper as it is difficult to compare it fairly to previous QDMR parsers.
>
> “the contribution from the NN architecture and the additional benefit from the DG signal”:
> Thanks for this question!
>
> We note that without the graph loss the relation embeddings don’t reflect the graph structure, which is what we are looking for.
> To tease apart the contribution of the architecture and the loss, we conducted an experiment that omits the loss function but instead uses embeddings of gold relations at *test time*. This led to very high accuracy (0.759 LF-EM) which shows that RAT-SQL is a good architecture for this task, but the challenge is learning the relations. We omitted this from the paper for brevity but will include it in the final version.
>
> “We now describe the details of the LF→DG conversion”
> The details were moved to the noted appendices, we will remove that line. Thanks!
>
> “ the cases in which LF-EM gets higher coverage than NormEM”
> As both methods are based on exact-match evaluation of long phrases, it is hard to classify the exact gap. LF-EM uses a wider “synonym vocabulary”, and defines a rich dedicated properties set (s.a., “more-than-1” of BOOLEAN, see Table 5 in the appendices) - thus catches more syntactic equivalences. Moreover, it declares role-based arguments names, and so agnostic to equivalent arguments ordering (e.g., ARITHMETIC[sum](arg=#1, arg=#2) ⇔ ARITHMETIC[sum](arg=#2, arg=#1); but  ARITHMETIC[div](left=#1, right=#2) != ARITHMETIC[div](left=#2, right=#1) ).

---

> > ### Comment · Reviewer_NVb5 · 2021-07-30
> > **Thank you for the response**
> >
> > Thanks for your clarifications and answers to my questions. Including response details in the paper will be very useful to readers IMO.

---

### Decision · Program_Chairs · 2021-08-17

**Decision:**

Accept

**Comment:**

This paper presents a method for QDMR parsing as dependency parsing. QDMRs are converted into dependency graphs, then a graph-based dependency parser is trained to produce these. Rather than just using the output graphs directly, they can be used in a RAT-based model which augments a seq2seq model with their information. The evaluation compares directly against seq2seq QDMR parsing models using an intermediate logical form representation (different from the dependency graph) and shows better results in out-of-domain QDMR parsing, but similar results in-domain compared to CopyNet+BERT. The reviewers liked the representation and found the overall quality of the experimentation and presentation to be high. A few concerns about clarity were raised and the model doesn't advance the in-domain state-of-the-art on this task, but these do not seem to be critical issues.